# Enhancement of Anticipatory Postural Adjustments by Virtual Reality in Older Adults with Cognitive and Motor Deficits: A Randomised Trial

**DOI:** 10.3390/geriatrics6030072

**Published:** 2021-07-22

**Authors:** Julien Bourrelier, Lilian Fautrelle, Etienne Haratyk, Patrick Manckoundia, Frédéric Mérienne, France Mourey, Alexandre Kubicki

**Affiliations:** 1Laboratoire INSERM U1093 CAPS, Université de Bourgogne Franche-Comté, 21078 Dijon, France; julien-bourrelier@gmail.com (J.B.); Patrick.manckoundia@chu-dijon.fr (P.M.); France.Mourey@u-bourgogne.fr (F.M.); 2Laboratoire ToNIC, Toulouse NeuroImaging Center, UMR1214, Inserm, UPS, 31000 Toulouse, France; lfautrelle@gmail.com; 3Equipe de Recherche Interdisciplinaire en Activités Physiques, Institut National Universitaire Champollion, STAPS, Campus de Rodez, 12000 Rodez, France; 4UFR Sciences et Techniques des Activités Physiques et Sportives (STAPS), Université de Bourgogne Franche-Comté, 21000 Dijon, France; etienne.haratik@gmail.com; 5Centre Hospitalier Universitaire, Pôle Personnes Âgées, 21000 Dijon, France; 6LE2I FRE 2005, Arts et Métiers, CNRS, UBFC, HeSam, 71100 Chalon-sur-Saône, France; frederic.merienne@ensam.eu; 7IFMS NFC, Hôpital Nord Franche-Comté, 25200 Montbéliard, France

**Keywords:** cognitive and motor deficits, postural control, rehabilitation exercise, virtual reality

## Abstract

Background: Postural activities involved in balance control integrate the anticipatory postural adjustments (APA) that stabilize balance and posture, facilitating arm movements and walking initiation and allowing an optimal coordination between posture and movement. Several studies reported the significant benefits of virtual reality (VR) exercises in frail older adults to decrease the anxiety of falling and to induce improvements in behavioural and cognitive abilities in rehabilitation processes. The aim of this study was thus to test the efficiency of a VR system on the enhancement of the APA period, compared to the use of a Nintendo Wii system. Methods: Frail older adults (*n* = 37) were included in this study who were randomized and divided into a VR exercises group (VR group) or a control group using the Nintendo Wii system (CTRL group). Finally, 22 patients were included in the data treatment. APA were studied through muscular activation timings measured with electromyographic activities. The functional reach test, the gait speed, and the time up and go were also evaluated before and after a 3-week training phase. Results and discussion: As the main results, the training phase with VR improved the APA and the functional reach test score along the antero-posterior axis. Together, these results highlight the ability of a VR training phase to induce neuromuscular adaptations during the APA period in frail older adults. Then, it underlines the effective transfer from learning carried out during the VR training movements to control balance abilities in a more daily life context.

## 1. Introduction

The frailty state is associated with a decrease in physical and/or cognitive abilities, functional reserves, and resistance to stressor events, which can lead to increased physical inactivity, disability, biological disturbances, risk of fall, and hospitalizations [1,2]. The diagnosis of frailty encompasses several domains, including weight loss, physical activity loss, a persistent state of fatigue, and physical impairments (grip strength, gait speed) [3]. In the context of geriatric rehabilitation, aged adults are admitted though multiple causes and contributors, such as malnutrition and clinical evidence of medical or psychiatric decompensation related or not with neurodegenerative disorders. By way of indication, dementia affects 17.8% of older adults aged over 75 years and 66% of people institutionalized [4,5]. Besides deficits in cognition, also balance, gait, movement control, and coordination are impaired in older adults with mild dementia and/or mild cognitive impairments and several diseases related to the advancement of age [6,7,8]. There is considerable evidence that deficits in balance and gait abilities can have consequences on the functional activities of daily life associated with an increased risk of fall and risk of becoming dependent [9,10].

Balance and postural control declines with age [11], especially in the case of associated cognitive impairments [12]. Postural activities involved in balance control integrate the anticipatory postural adjustments (APA) that stabilize balance and posture [13], facilitating arm movements and walking initiation [14,15] and allowing an optimal coordination between posture and movement [16]. Several studies highlighted the disturbances associated with age or diseases on these crucial components of postural and balance control. It is widely acknowledged that the balance training used in preventative care actions are able to maintain and/or strengthen the postural abilities in aged adults [17], especially the APA efficiency [18]. Moreover, this kind of exercise therapy also works in frail aged adults: several studies showed improvements in anticipated postural control using ball exercises [19] or playing with serious games to enhance the patients’ motivation to move faster [20].

Serious games, and more recently even virtual reality (VR) games, are often used to improve the rehabilitation environment, offering multiple possibilities to assess and stimulate functional abilities [21]. VR can provide a medium for placing the patient in an ecological environment for performing balance and gait exercises [22]. Virtual environments can be programmed to reproduce the realistic stimuli in order to activate the exact cognitive and motor behaviours associated with the environment with respect to the participant’s abilities and the rehabilitation objectives. The immersive degree plays a crucial role in the representation of real-world situations, allowing for the assessment of the real behaviour of patients in front of stimuli [23].

Starting from this postulate, several studies showed the potential of the VR used in the gait and balance program in frail older adults. The Nintendo Wii system has been widely studied in a geriatric context in order to encourage the movement and entertainment of aged adults. These studies showed the relative acceptability of these technologies and the voluntary engagement in games according to new cognitive and motor interactions, with new environments. They also noted improvements in muscular function, measured by maximal voluntary contraction, or balance function, measured by the Timed Up and Go or Chair Stand tests [24]. All of these studies attested to the learning abilities and promising results in frail aged patients during and after stimulation programs. Unfortunately, to our knowledge, none of these studies measured the gain in terms of APA. Nonetheless, neither study highlighted a virtual environment providing an accurate APA stimulation in the context of a gait and balance rehabilitation program. To train APA, it seems important to stimulate fast movements, and to give almost at the last second the exact location of the target [18,20].

The aim of this study is to test the efficiency of a VR system developed with physical therapists in the enhancement of the APA period, compared to the use of a Nintendo Wii system. Consequently, we hypothesized that the group using this VR system will show better APA improvements compared with the group using the Nintendo Wii system.

## 2. Material and Methods

### 2.1. Participants

Frail older adults were included at the Geriatric Medicine and rehabilitation service of Dijon University Hospital, France. All the participants were right handed, as assessed by the Edinburgh Handedness Inventory [25], and gave written informed consent, in accordance with the local ethical committee clearance ID-RCB number 2015-A01583-46, Clinical Trial Reference number NCT02855853. This work was carried out in agreement with the Code of Ethics of the World Medical Association (Declaration of Helsinki) for experiments involving humans. The privacy rights of human participants were observed at all times. Inclusion criteria were (1) recognized cognitive deficit with ability to understand simple orders according to a Mini Mental State Examination [26] (MMSE) scored between 15 and 28; (2) recognized motor deficit highlighted by a gait velocity lower than 0.65 m.s^−1^ [27]; (3) performed the 10 sessions of the rehabilitation program in 3 weeks; and (4) realized the pre- and post-test sessions. Exclusion criteria were (1) a non-affiliation to a social security scheme; (2) an insufficient cognitive state (MMSE score < 15); (3) an insufficient motor state (inability to walk more than 10 m, and/or inability to maintain the upright standing posture for more than 60 s); (4) severe disorders of vision and/or audition (inability to perceive visual and/or auditory information coming from virtual reality); (5) a cardiovascular affection in the acute or unstable phase; (6) an unbound bone fracture; (7) an interruption of over 96 h between the two rehabilitation sessions; (8) a missed a rehabilitation session; and (9) failed to attend the pre- or post-test session. To comply with these inclusion criteria, a sample size of 20 patients should be recruited. 

For the present study, 637 patients were approached (see the enrolment and allocation stages in Figure 1). Among them and according to the inclusion/exclusion criteria, 528 patients could not be included, 7 patients did not consent to participate, and so 37 patients were finally included in the present clinical experimental protocol. These 37 patients were randomized and divided into a virtual reality exercises group (VR group) or a control group using the Nintendo Wii system (CTRL group). A clinical research associate was assigned to enrol participants and assign them to the interventions. Fifteen patients had to be excluded during the experiment due to a missing rehabilitation session (*n* < 10 sessions in 3 weeks) or an alteration in the general state of their health (for example: a cardiovascular complication; *n* = 5). Finally, *n* = 22 frail aged patients achieved this experimental clinical protocol in the VR (*n* = 11) and CTRL (*n* = 11) groups.

### 2.2. Experimental Design

The full experimental design and procedure are illustrated in Figure 2. The experiment consisted of a pre-test session during which the MMSE, 10 m gait speed, Timed Up and Go test [28] (TUG), and functional reach test [29] (FRT) were recorded. Moreover, consistent with our previous studies [30], an arm-raising protocol was conducted while finger kinematics and surface electromyographic activities (EMG) from 11 muscles were recorded in order to evaluate the motor programs and the APA [31] (please see the arm-raising procedure and devices paragraph). Then a rehabilitation training phase began within 3 days after this pre-test session. This training phase consisted of 10 sessions in three weeks with a total of 5 sessions per week, during which the patients either performed games in a virtual reality environment (VR group) or performed bowling on video games (CTRL group). Finally, after a 24 h recovery period after the last training session, the post-test session was performed. In this post-test session, exactly the same procedures and apparatus that had been used in the pre-test session were repeated.

### 2.3. Arm-Raising Procedure and Devices

Before the arm-raising protocol, every patient performed a short warm-up of the shoulder muscles. The arm-raising task was exactly the same as in our previous published work [30]. In summary, patients were instructed to start from a standardized standing posture (Figure 2, panel 1) and asked to point their index finger towards a left or right diode as soon as the light switched on. These two diodes were fixed 2 m high from the ground, 2.5 m in front of the participant’s feet, and 1.2 m apart, with a central point between them exactly in front of the patients’ right shoulder. Participants were unaware of the visual stimuli location (left or right). They were instructed to raise their arm as quickly as possible after the appearance of the visual stimuli. This complex reaction time task with a relative location uncertainty generates greater APA than a simple reaction time task [18]. The arm-raising protocol in the pre- and post-test sessions were composed of 20 pointing trials while movement kinematic and surface EMG activities from 11 muscles were recorded. Hand kinematics were recorded using the VICON system (Oxford metrics group, UK) at a sample frequency of 100 Hz. Reflective passive kinematic markers were placed on the right upper arm on the following anatomical loci: nail of the index finger, dorsal aspect of the scaphoid bone, lateral aspect of the elbow, and anterior aspect of the acromion. The three-dimensional path coordinates of the patients’ arm markers were captured, reconstructed, and labelled by the Vicon software, following which smoothing was performed using cross-validation splines [32]. Trajectory data were filtered using a fourth-order Butterworth filter with zero lag and a cut-off frequency of 30 Hz. From these kinematic data, the finger movement onsets were defined when the linear tangential velocity of the index fingertip was equal to 5% of the maximal velocity before the peak value was reached for every trial in every patient. Such a 5% threshold allowed for the detection of the finger movement onset without false-positive detection using an automatic procedure with MATLAB scripts (MATLAB R2014b, Mathworks). The finger reaction time was then calculated as the time-lag between the light-on of the visual target and the finger movement onset. Surface EMG activities (ZERO WIRE EMG system, Aurion, Italia) of 11 muscles were also recorded at a sampling frequency of 2000 Hz and synchronized with the kinematic recordings. According to the EMG literature concerning the APA [33], the EMG recordings are focused on the lower limbs and the trunk muscles in reference to the finger movement onset. EMG was recorded in a bilateral manner on ten muscles: the right and left rectus femoris (rRF, lRF), biceps femoris (rBF, lBF), obliquus internus (rOI, lOI), erector spinae at the level of the third lumbar vertebra (rESL3, lESL3), and the erector spinae at the level of the seventh dorsal vertebra (lESD7, rESL3). Moreover, the right anterior portion of the deltoidus (rDA) was also recorded (Figure 3A). Patients were instructed how to selectively activate each recorded muscle individually [34] in order to determine the positions of the surface electrodes. These electrodes were placed parallel to the muscle fibres with an inter-electrode distance of 2.4 cm. The overall EMG signals were first bandpass filtered between 5 and 400 Hz, full-wave rectified, and then filtered using a no-lag averaging moving-window algorithm with a 10 ms window size. The procedure for setting the timing of the muscle recruitment was performed for each muscle recording of each participant individually. For each muscle, EMG signals were thus first integrated per 10 ms interval (iEMG) from the illumination of the target diode (i.e., when the voluntary movement was coming), and during the EMG baseline between −2 to −1 s before the diode was lit (i.e., during maintenance of the initial position). Then we compared at each time point these iEMG values between these two EMG signals using t-tests to determine the timing of the muscle recruitment. The first instant at which the *p*-value was lower than 0.05 for a duration of at least 50 ms determined the beginning of the muscle activation necessary to perform the arm-raising pointing movement. It is very important to note that repetitive t-tests were not used here to answer the research question as to whether there is a difference or not between the two EMG signals (one voluntary movement versus homeostatic initial position, we know that there will be a difference). Repetitive t-tests were used here to answer the research question when the difference between these two EMG signals occurred and was significant for at least 50 ms. In this precise case and with this methodology, there was no summation of the false-positive rate. Certainly, this difference was subtle but legitimized the use of repetitive t-tests in the design of our method of signal processing from a methodological point of view. This method has been previously tested and used in several studies regarding arm raising or pointing movements [31,35,36,37,38,39,40,41]. As defined in our previous work [31], the beginning of significant muscle activation was defined as the initiation time. To compute these initiation times for the 11 recorded muscles, the EMG values after the illumination of the diode and the EMG baselines were compared for each value using t-tests for each muscle in each group. For each patient, the mean integrated activity of each muscle from −2 to −1 s before the first diode was lit during the maintenance of the initial position was defined as the muscle EMG baseline. The first instant at which the *p*-value was lower than 0.05 for a duration of at least 50 ms determined the beginning of the muscle activation necessary to perform the arm-raising pointing movement. In every movement, the timing of each muscle activation was calculated in each patient for each muscle recording prior to finger movement onset. This parameter was named the activation timing. Consequently, the activation timing of the muscles involved prior to the finger movement onset had negative values (Figure 3B, right column), whereas the activation timing of the muscles with positive values refer to post finger movement onset muscular activations (Figure 3B, left column).

### 2.4. Training Phase

The duration of each training session in the two groups (VR and CTRL) was similar and approximately 30 min for 50 arm movements in total.-The VR group: In order to propose rehabilitation exercises potentially increasing the management of balance abilities, a VR environment was designed using a visual immersion system, Cave Automatic Virtual Environment (CAVE). This environment is made up of two screens (resolution of 1024 by 768 pixels): (i) the first was a front wall measuring 2.70 m high and 3.40 m wide; and (ii) the second was a floor measuring 3 m depth and 3.40 m wide (Figure 2, panel 2). The interactivity between patients and the virtual environment was supported by two complementary technologies: (i) active stereoscopic vision (NVidia 3D Vision Pro) with special 3D glasses; and (ii) a tracking system composed of 4 infrared cameras (ART DTrack 2) operating at 60 frames per second with a precision of 1 mm. Two parts of the patients’ body are tracked with markers, which allowed an egocentric interaction in the virtual environment. Markers are positioned on the 3D glasses to capture the movements of the participant’s head. Other markers were placed on a wand, which is gripped by the participant to capture the movement of the dominant arm. Participants stood 3 m in front of the front wall screen. Patients received as instruction: “please pick the red apples”. Two different scenarios of the rehabilitation exercises were realized. (A) In the first scenario, called “picking ripe apples”, patients had to perform quite fast arm movements toward the virtual apples (apples played the role of targets). These apples had dynamic changing states: they appeared for the first time in the form of an apple blossom, became green, yellow, and then red, and finally were black apples. The changing state velocity was pseudo-randomized within a 500 to 5000 ms window depending on the patients’ abilities. These pseudo-randomized velocities were chosen by the medical staff during each training session. Once caught, the red apple had to be put in a real wicker basket which was integrated into the action space of the virtual environment in order to reinforce the ecological aspect of the motor tasks. Before each training exercise, the medical staff could adjust several other parameters accordingly to the patients’ state of health on the day: number of apples by exercise, each apples’ positions, the apples’ sizes, and the wicker basket position could be modulated. Overall, from these parameter modulations, the medical staff were able to induce an increase in the arm-raising velocities and thus biomechanically increase the difficulty level of the motor tasks with regard to equilibrium management for the patients. (B) The second scenario was called “the successful picking” and proposed in a virtual environment an apple tree with several red apples that had to be picked and put in the same wicker basket as used in the first scenario until a visual time gauge expired. Once again, medical staff could adjust several parameters in this motor exercise. Using both of these scenarios, the medical staff designed and regulated each training session in order to allow patients to perform 40 arm movements.-The CTRL group: From an upright standing position, patients played two parts of Wii Sport Bowling (Nintendo Wii), inducing a minimum of 50 arm movements. Such a protocol was systematically proposed for all patients in the Geriatric Medicine and rehabilitation service of Dijon University Hospital, France, in which this clinical research protocol was conducted. For ethical reasons, it was not possible to offer fewer motor stimulations to patients included in this study than the usual care protocols. This is why this group included the control group in the present clinical study.

### 2.5. Statistical Analyses

First of all, the data coming from the clinical tests (Gait Speed, TUG, and FRT), the finger reaction times, finger velocity peaks, and all the activation timings coming from the EMG data were checked for a normal distribution using Shapiro–Wilk W tests (all the *p*-values were >0.12). The variances in each variable were also assessed for equality using Levene’s test (all the *p*-values were >0.4). The effect of the rehabilitation protocol on clinical tests, motor control, and more precisely management of balance abilities was assessed by comparing the data coming from the pre-test session with those obtained in the post-test session phase. The clinical test results (Gait Speed, TUG, and FRT), finger reaction times, and finger velocity peaks were analysed with a mixed ANOVA. Test session (pre-test/post-test session) were within-patient measures while the rehabilitation training group (VR/CTRL) was a between-patient factor. Tukey HSD post-hoc tests were carried out where appropriate. Moreover, according to Cohen, the effect size was specified by the partial eta squared (η^2^) [42]. Concerning the EMG data, the four average sequences of muscular activation timings were computed for the pre- and post-test sessions in the VR and CTRL groups. These four whole activation timing sequences were compared by pairs with linear correlations. It is important to note that the linear correlations were used here as a measure of similarity and for comparison between groups and time points. Moreover, lESl3 and rRF activation timings present a great interest: firstly, these postural muscles are mainly involved in the management of balance during arm raising and APA [33]. Secondly, our previous work highlighted significant modulations of the activation timings in these muscles when patients suffering from mild cognitive impairments performed arm-raising movements [30]. For this reason, the activation timings of the lESl3 and rRF were assessed using a similar mixed ANOVA as described above. From a statistical point of view, such analyses could be interpreted as multiple tests of significance on the same EMG dataset. For this reason, a Bonferroni correction with a level 2 (left ESl3 and right RF) was applied. For the sake of clarity throughout this article, the corrected *p*-values were presented when dealing with the results of the ANOVA performed on the activation timings of the left ESl3 and the right RF. Once again, Tukey HSD post-hoc tests were carried out where appropriate.

## 3. Results

To achieve the main aim of this study, the effects of the virtual reality training were assessed by comparing the post-training phase results with those obtained in the pre-training phase.

In relation to the finger reaction times, the ANOVA revealed no session (F(1,12) = 0.860, *p* = 0.37), no group (F(1,12) = 0.310, *p* = 0.59), and no interaction effects (F(1,12) < 0.01, *p* = 0.99) (Figure 4 left column). Regarding the finger velocity peaks, the ANOVA revealed no group effect (F(1,12) = 1.63, *p* = 0.226), a significant session effect (F(1,12) = 12.67, *p* < 0.01, η^2^ = 0.52), as well as a significant interaction of training group x session effect (F(1,12) = 19.79, *p* < 0.001, η^2^ = 0.62) (Figure 4, right part). More precisely, the Tukey HSD post-hoc tests highlighted faster finger velocity peaks in the post-test phase solely for the VR group (*p* < 0.001). Interestingly, the finger velocity peaks in the post-test phase were significantly faster in the VR group than in the CTRL group (*p* < 0.01).

Figure 5 shows the average activation timing of whole sequences in the pre- and post-test phase for the VR and the CTRL group. As previously mentioned, linear correlations were used as a measure of similarity and for comparison between groups and time points. During the pre-test session and for the 10 bilateral recorded muscles, no initiation time appeared before the finger movement onset. To compare these four average activation timing sequences, the correlation coefficient R between pairs of conditions were computed and the *p*-value was reported according to the Pearson’s Correlation Table. The average activation timing sequence in the pre-test phase between the VR and the CTRL group of the patients were strongly correlated (R = 0.98, *p* < 0.001). Only the sequence of the average activation timing recorded during the post-test phase in the VR group showed a clear difference and no significant correlation in the pre-test phase in the same group (R = 0.54, *p* > 0.05) nor in the post-test phase of the CTRL group (R = 0.52, *p* > 0.05).

As explained in the statistical analyses paragraph of the material and methods section, based on our previous works and the APA literature, we hypothesized that the left ESl3 and the right RF activation timings could be modulated and shortened with the VR training. To test these hypotheses, two repeated mixed ANOVA with appropriate Bonferroni correction were conducted (see the statistical analyses section). These results are presented in Figure 6. Regarding the left ESl3, the ANOVA revealed the main effects of the group (F(1,12) = 8.13, *p* < 0.01, η^2^= 0.40), session (F(1,12) = 12.59, *p* < 0.01, η^2^ = 0.52), as well as a significant interaction of group x session effect (F(1,12) = 4.90, *p* < 0.05, η^2^ = 0.30). The Tukey HSD post-hoc tests revealed that the activation timings of the left ESl3 were significantly shorter in the post-test phase for the VR group only. More precisely, in the post-test phase for the VR group, the activation timings of the left ESl3 (−0.062 ± 0.048 s in average) occurred before the finger movement onset, and were (i) significantly shorter than during the pre-test phase in the VR group (0.110 ± 0.091 s in average; *p* < 0.01); and (ii) significantly shorter than during the pre-test and post-test phase in the CTRL group (respectively, 0.160 ± 0.103 and 0.120 ± 0.100 s in average; ps < 0.01). Regarding the mean evolution of the activation timings in the left ESl3 from the pre-test to the post-test phase in the CTRL group, the Tukey HSD post-hoc tests revealed no significant difference (*p* = 0.77). Regarding the right RF, the ANOVA revealed no session (F(1,12) = 1.18, *p* = 0.30), no group (F(1, 12) = 0.79, *p* = 0.40), and no interaction effects (F(1, 12) = 0.10, *p* = 0.75).

Then, we considered the scores of the clinical tests obtained during the pre- and post-test phases. These results are represented in Figure 7. Regarding the FRT, the ANOVA revealed no significant effect of group (F(1,20) = 1.50, *p* = 0.24), a significant effect of session (F(1,20) = 7.05, *p* < 0.05, η^2^ = 0.26), as well as a significant interaction of group x session effect (F(1,20) = 8.85, *p* < 0.01, η^2^ = 0.31). The Tukey HSD post-hoc test revealed that the FRT scores increased on average during the post-test phase only for the VR training group (from 16.3 ± 7 cm to 21.7 ± 5 cm in average, *p* < 0.001). These average FRT scores during the post-test phase in the VR group were significantly better than those in the CTRL group (17.9 ± 4 cm, *p* < 0.05). No significant effect was revealed by the Tukey HSD post-hoc analysis in the CTRL group between the pre- and post-test phases (*p* = 0.88).

Regarding the mean gait speed over 10 m, the ANOVA revealed both a training group effect (F(1,20) = 4.67, *p* < 0.05, η^2^ = 0.18) and a session effect (F(1,20) = 6.65, *p* < 0.05, η^2^ = 0.24) but no interaction effect (F(1,20) = 0.02, *p* = 0.96). The Tukey HSD post-hoc revealed an increase of the mean gait speed from the pre-test to the post-test phase in the two groups (from 0.37 ± 0.15 m.s^−1^ to 0.57 ± 0.24 m.s^−1^, *p* < 0.05 in the VR group; from 0.60 ± 0.32 m.s^−1^ to 0.82 ± 0.28 m.s^−1^, *p* < 0.05 in the CTRL group). No significant difference was detected during the post-test phase between the VR and CTRL group (*p* = 0.30).

Finally, no significant effect was detected for the TUG scores, which were on average 25.9 ± 9.2 and 24.3 ± 8.6 s in the pre- and post-test phase, respectively, for the VR group, and 16.3 ± 5.9 and 17.1 ± 4.5 s in the pre- and post-test phase, respectively, for the CTRL group.

## 4. Discussion

In this study, we hypothesized that muscular synergy linked with arm-raising movements can be modified after specific virtual reality training in frail aged adults.

After three training weeks, the results showed significant changes in the VR group, compared with the sham Wii group. Indeed, we noted an increase in finger velocity and a variation in the associated muscular synergy in the VR group: lESl3 is activated earlier in post-training tests. It is worth noting here that instructions were given to reach the apple as fast as possible in the VR group. To the contrary, the Wii exercises do not require quick motor responses. The start and the velocity of movement is in self-paced mode compared to the VR exercise where the rhythm request is imposed by features of the exercises and so by the therapist. Considering this, it seems appropriate to note that only the participants of the VR group increased their movements’ velocity. The VR software has been specially developed to involve wide and rapid movements; it is probable that this game specificity is sufficient to explain the velocity increase. Previous studies already highlighted velocity gains during arm movements in aged adults [18,43] and in frail aged adults [20]. However, as expected in the hypothesis, we also note a substantial variation in the muscular synergy associated with the arm movement itself. The lESl3, a key muscle involved in the anticipatory postural adjustments [44], is recruited earlier in the post-test. As APA are also intended to facilitate and optimize movement itself [16], the early activation of this postural muscle is probably another reason to explain the increased finger velocity. In the subjects of the VR group, coordination between posture and movement has changed over ten training sessions in three weeks, revealing a probable optimized APA period to facilitate arm movement and stabilize posture. This result is encouraging for rehabilitation processes with these patients.

It is questionable why the Wii program seems unable to optimize the coordination between posture and movement in the tested patients, whereas many previous studies showed potential interests in clinical practice [45]. However, we identified several differences between the VR tasks and the Wii bowling simulation that could explain this. First of all, as explained above, the VR movements were wider and faster than those involved in the Wii video game, leading to more inertial charges that the CNS has to manage in order to maintain balance. Interestingly, we can put these results in perspective with precedent papers from other teams, showing optimized APA after training based on ball passing with different loads [46]. In their study, Aruin and colleagues trained older adults to self-initiate throwing movements with a medicine ball. This kind of training with wide movements combined with heavy inertial charges induced significant early activation of the recorded postural muscles; in other words, an improvement of postural control and functional balance. In this way, it seems important to work with fast, wide, and relatively unpredictable movements to improve the APA efficiency. Secondly, in the VR program, we designed targeted movements in which the hand of the participants had to reach a precise spatial location to succeed the task. In the Wii bowling simulation, the movement is more intransitive, and the task can be successful or missed for the same final position of the hand. In this way, it seems important to work with transitive and targeted movements to improve the APA efficiency. Such conclusions about targeted movements and APA improvements were also recently shared in stroke patients [47].

Therefore, in the context of balance rehabilitation, we suggest creating a stimulating environment to compel the patient to go “beyond their comforts zones” in respect of security and the achievement of the set goals in a proposed situation.

One may ask why only lESl3 is recruited earlier when APA are optimized. Indeed, several other muscles (such as the right and left biceps femoris of the right erector spinae at L3 or erector spinae at D7) are involved in the APA period. In particular, it could be intended to facilitate an early activation of the right rectus femoris, considering its early recruitment into the muscular chain. Nevertheless, our training significantly revised only one muscle timing (lESl3) and not the others. It may be necessary to train longer (only three weeks here), with longer sessions (only 50 movements during 30 min here), or to solicit arm movements with a more important lateral component to see a more global modification of the muscular synergy. Another possibility is that inferior limbs were not sufficiently involved in the proposed exercises. These training conditions should be used in future research.

Surprisingly, reaction times were not reduced in the participants of the VR group. As these participants have to reach quickly when the red apple appears, we hypothesized an improvement in reaction times. We can speculate that the start of the movement was too difficult to detect for the participant.

The results also show several interesting points in view of clinical transferability: gait speed improves in both groups, which means that the two training sessions were efficient to challenge the whole motor function and sufficient to involve an increase in gait speed. As none of the games directly solicited inferior limbs movements, this result is interesting. We could speculate that this change could be due to the motor activity involved in each training protocol, or by contextual effects [48], since the participants of both groups were included in a rehabilitation program. A sham group with no stimulation would allow for this hypothesis to be verified in further studies. Moreover, the Functional Reach Test increased only in the VR group. This test represents an interesting outcome in aged adults because it correlates with several aspects of daily living independency [49,50]. During the FR test, the subject is asked to move forward at a natural speed; it is not a rapid movement. Then, the learning effect seen in the participant of the VR group, trained with wide and fast movements, seems to be transferred for wide but slow movements, such as the FRT movement. Interestingly, the FRT scores are correlated with the fear of falling in aged adults [51]. Fear of falling is very common in frail patients, and can be reduced by specific balance training [52] and VR training involving balance control and upper limb movements [53].

This study promotes the use of virtual reality in clinical rehabilitation with frail aged adults. Several research studies showed the feasibility of this approach with aged adults, with suitable conditions [54] and in a safe environment [55], promoting an improvement in their motivation [56] and optimizing their rehabilitation outcomes [57]. These technologies represent an interesting tool for therapists who can easily tailor and grade the exercises according to the patient’s needs and abilities. Specifically, in the context of postural control in older adults, the features of the environment were defined by the number of targets; the nature, size, and colour of the targets; the minimal and maximal height; and the maximal reach distance of the patient.

We should acknowledge that there were some limitations concerning this study. First of all, the small number of participants does not facilitate a generalization of the results, which could be reached in a more important randomized control trial. Another limitation is the absence of a multi-criteria frailty diagnosis. The gait speed was mainly used to detect a kind of frailty (based on the gait speed) in this paper, which allowed us to detect a motor deficit but not a real frailty state. In further studies, other signs should complete the clinical examination in order to confirm the diagnosis, such as the other Fried criteria. However, we tried to take into account the cognitive deficits, using the MMSE that is often associated with frailty.

We also have to underline the relatively large dropout rate, consistent with this frail population. A qualitative research approach could have been done to explain these behaviours. Further studies could be designed in a mixed approach.

From the observations made during the training of the frail older adults, we think that it is a powerful tool to improve motivation. It also improves the dose effect of exercises by increasing the repetition with a great possibility of training variations. We did not measure these important parameters; this should be done in further studies. However, the tools using these new technologies must take into account the postural and motor characteristics of older adults and be designed according to specific objectives.

## Figures and Tables

**Figure 1 geriatrics-06-00072-f001:**
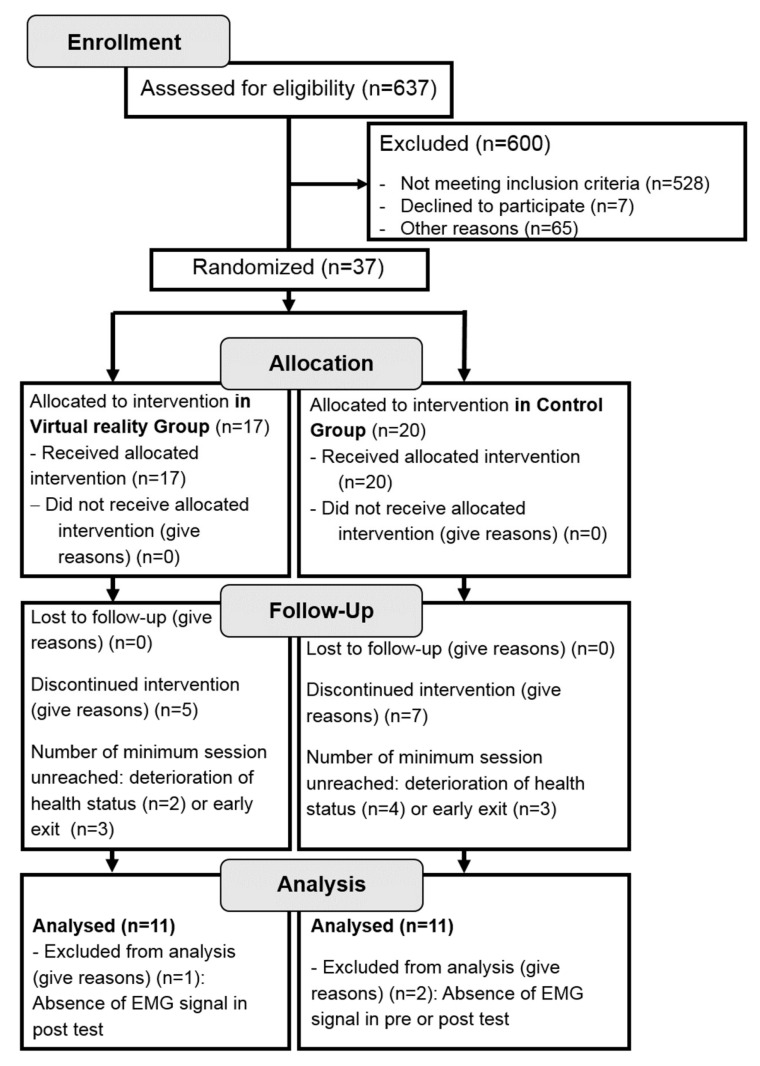
CONSORT flow chart of the study.

**Figure 2 geriatrics-06-00072-f002:**
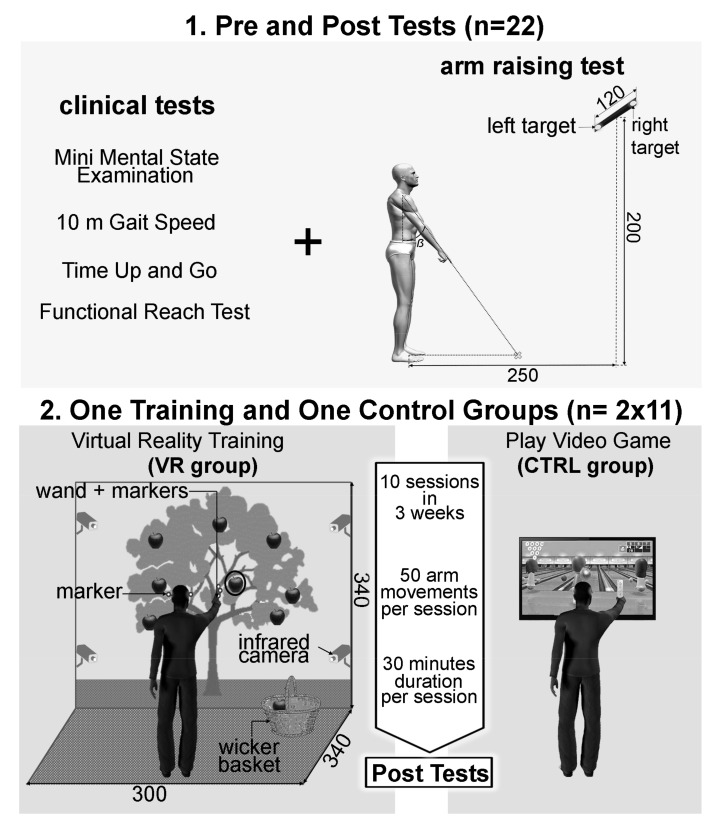
This figure summarizes the whole experimental design. Panel 1: overview of all the tests used to measure the rehabilitation training effects. On the left side of Panel 1 are the clinical tests used, and on the right of Panel 1 are the experimental set-up and apparatus of the arm-raising test used to evaluate the neural mechanisms of the motor control during voluntary arm-pointing movements. Here the participant maintains the standardized initial starting posture. The distances are reported in centimetres, and the ẞ angle measures 35°. Panel 2: overview of the whole rehabilitation training phase, for both experimental groups. On the left of Panel 2 are the virtual reality set-up and apparatus used for the VR group. As in Panel 1, distances and dimensions are reported in centimetres. On the right of Panel 2 is the training set-up with a ‘Nintendo Wii’ used for the CTRL group. Between these two halves of Panel 2, the duration of the rehabilitation training phase, the number of movements by session, and the duration per session are reported. These parameters of the rehabilitation training phase were similar for both the VR and CTRL groups.

**Figure 3 geriatrics-06-00072-f003:**
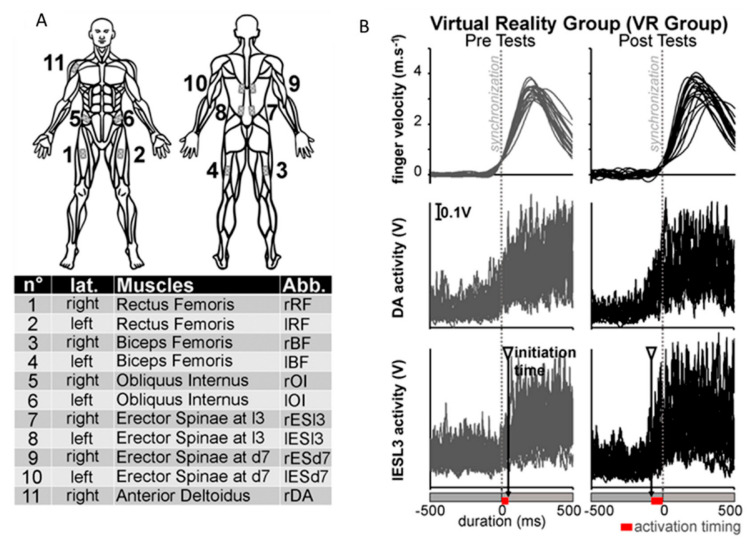
Panel (**A**): The upper part locates and numerates the recorded muscles on the anatomical representation of the human body. The lower part summarizes for each muscle number from left to right: the laterality, the muscle name, and its abbreviation in this article. Panel (**B**): Raw data of a typical participant from the VR rehabilitation training group. In each graph, the 20 recording trials are superimposed. The left column reports the pre-test measures with grey lines whereas the right column reports the post-test measures with black lines. All the recording data were synchronized to the hand movement onset and represented here from 500 ms before the hand movement onset to 500 ms after. In lines, from top to bottom, the finger velocities (m.s^−1^), the deltoid (anterior portion, DA), and the left erector spinae at the third lumbar vertebra (lESL3) are reported. Kinematics signals are filtered with a dual-pass algorithm with a 30 Hz cut-off frequency (Butterworth 4th order). EMG signals are bandpass filtered between 5 and 400 Hz, full-wave rectified, and then filtered using a no-lag averaging moving-window algorithm with a 10 ms window size. The following overall arm-raising movement analyses and more particularly the detection of initiation time in the EMG activities were carried out using this kind of typical data. For the lESL3 panels, in the pre- and post-tests, the average initiation time is highlighted using a grey triangle and a black vertical arrow. Consequently, along the duration scale, for both the pre- and post-tests, the average activation timing (i.e., the delay between the hand movement onset and the average beginning of the significant lESL3 activation) is highlighted by a red rectangle. (For interpretation of the references to colour in this figure legend, the reader is referred to the web version of this article).

**Figure 4 geriatrics-06-00072-f004:**
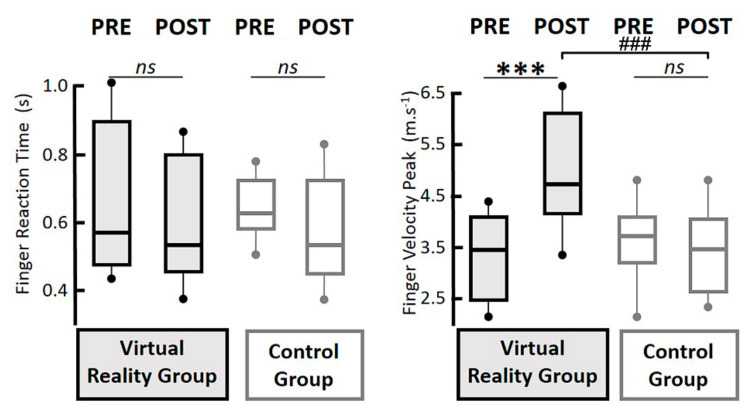
Boxplot graphs. In each reported parameter, the box and whiskers report the values of all the recordings for all participants. The box is defined by the first and third quartiles. The thick line within the box represents the median value. The tenth and ninetieth percentiles are plotted as circles above and below the whisker bars. All participants in the VR group are reported in black lines with a grey area, and the CTRL group in grey lines with a white area. In the left panel, the boxplots represent the finger reaction times in seconds. In the right panel, the boxplots represent the finger velocity peaks in m.s^−1^, according to the pre- and post-test sessions (upper line). ns, *p* > 0.05; ***, *p* < 0.001 between the pre- and post-tests within the VR group; ###, *p* < 0.001 between the post tests in the virtual reality group and the pre/post-tests in the control group.

**Figure 5 geriatrics-06-00072-f005:**
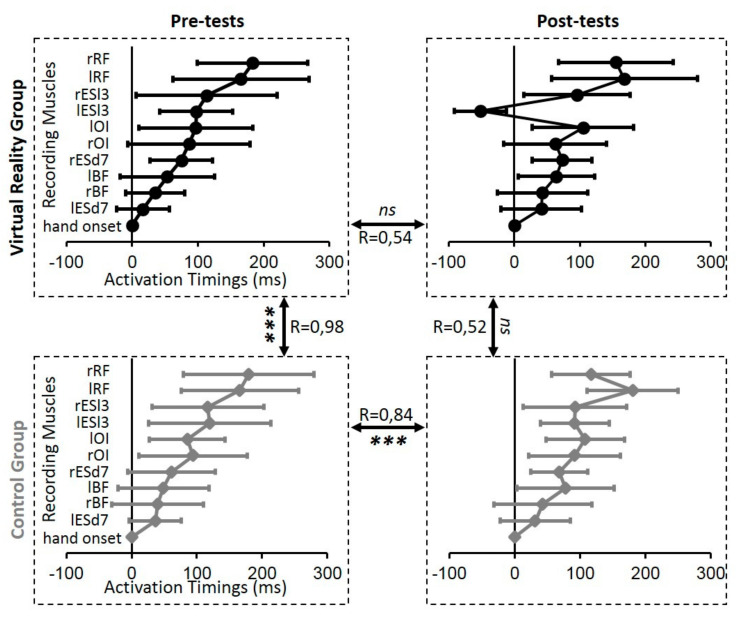
Whole sequences of averaged (± standard deviation) muscular activation timings (in milliseconds) used to initiate the arm-raising movement in the pre- and post-test sessions (in column) for all participants in the VR (upper line) and the CTRL (bottom line) groups. Regarding the duration axis, the origin (0) corresponds to the hand movement onset. To compare these four global sequences of muscular activation timings, the correlation coefficient R between each pair of whole sequences was computed and reported here between the relevant pair of graphs. *** *p* < 0.001.

**Figure 6 geriatrics-06-00072-f006:**
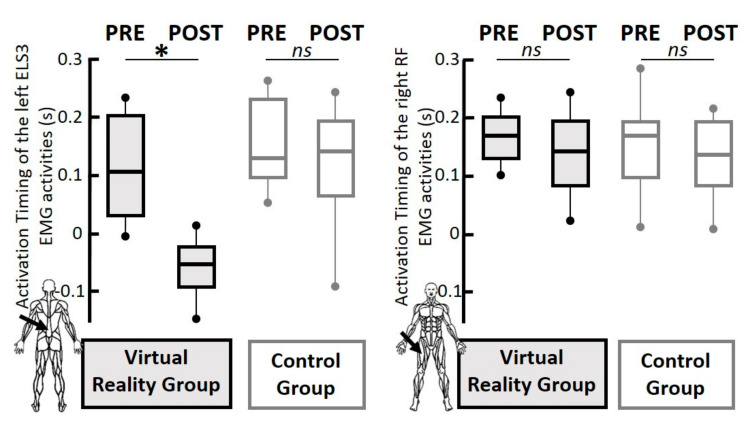
Boxplot graphs. In each reported parameter, the box and whiskers report the values of all the recordings for all participants. The box is defined by the first and third quartiles. The thick line within the box represents the median value. The tenth and ninetieth percentiles are plotted as circles above and below the whisker bars. All participants in the VR group are reported in black lines with a grey area, and the CTRL group in grey lines with a white area. Left panel: Boxplot graphs of the activation timing (second) of the left erector spinae at the third lumbar vertebra (lESL3) EMG activities. Right panel: Boxplot graphs of the right rectus femoris (rRF) EMG activities. The location of the concerned muscle is reported by a black arrow on a body schema in each graph. *ns*, *p* > 0.05; *, *p* < 0.05.

**Figure 7 geriatrics-06-00072-f007:**
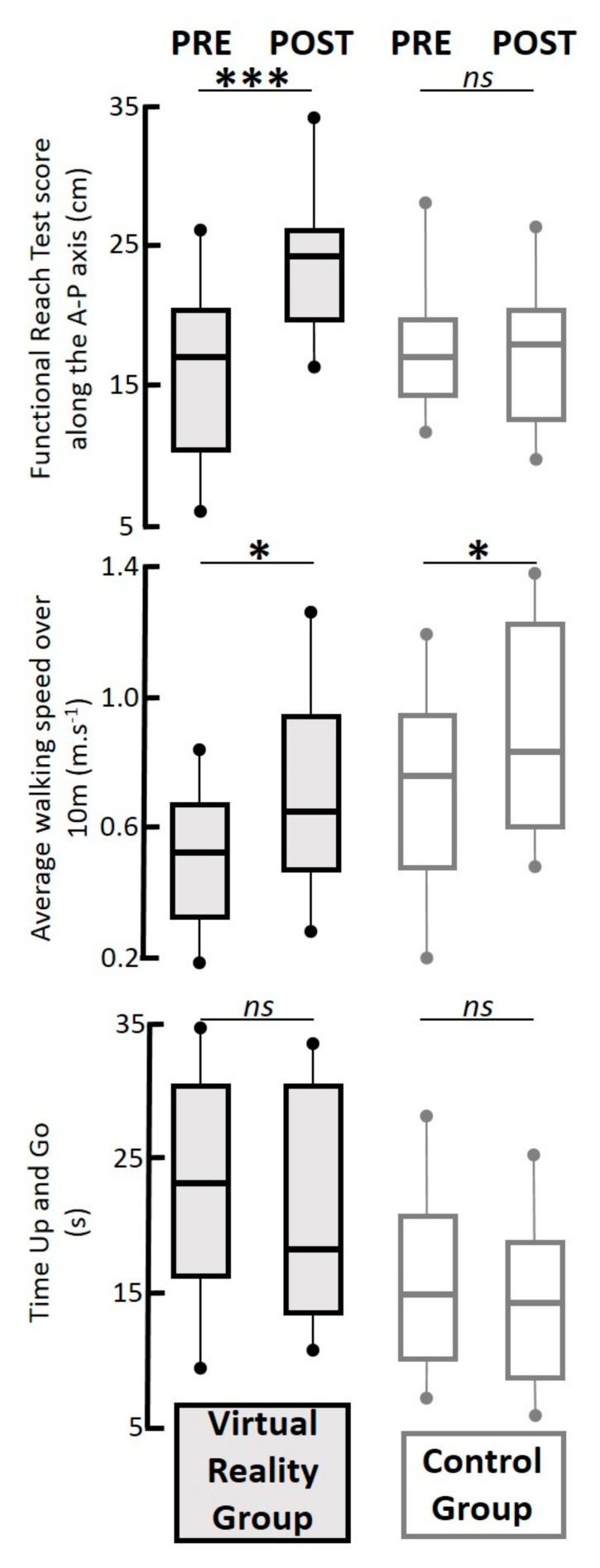
Boxplot graphs of the three clinical tests used to characterize the pre–post impacts of the rehabilitation programs (in column) in both groups (VR group with black lines and grey area; CTRL group with grey lines and white area). In each reported parameter, the box and whiskers report the values of all the recordings for all participants. The box is defined by the first and third quartiles. The thick line within the box represents the median value. The tenth and ninetieth percentiles are plotted as circles above and below the whisker bars. All participants in the VR group are reported in black lines with a grey area, and the CTRL group in grey lines with a white area. From top to bottom, the functional reach test score measured along the antero–posterior (A–P) axis in centimetres, the average walking speed over 10 m in m.s^−1^, and the Time Up and Go score in seconds. In each graph, the results are reported for all participants. *ns*, *p* > 0.05; *, *p* < 0.05; ***, *p* < 0.001.

## Data Availability

All the data are available by the backup process of the corresponding author.

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
