# Peer review of "Enhancement of Anticipatory Postural Adjustments by Virtual Reality in Older Adults with Cognitive and Motor Deficits: A Randomised Trial"

_geriatrics, 2021, doi:10.3390/geriatrics6030072_

Round 1
Reviewer 1 Report
Review: Enhancement of anticipatory postural adjustments support by virtual reality in frail elderly patient(s)
General comment
The paper tests a virtual reality-based intervention to improve postural function during reaching arm movements in a frail elderly group. Several functional tests were carried out pre and post-intervention, while muscle activation timings were used to reveal changes in anticipatory postural adjustments. The study's findings led authors to conclude that virtual reality training is efficacious in improving anticipatory postural adjustments in relatively short time in frail elders compared to a Nintendo Wii task. The study is a relevant contribution to improving rehabilitation and training of postural function. However, there are some questions and issues to be addressed.
Abstract
- The abstract could be improved and made more accurate. For instance, the sentence "Several studies showed the potential of Virtual Reality (VR) used in ..." does not tell what the "potential" is.
- Also, the sentence: "Patients were evaluated...though the anticipatory postural adjustments (APA)..." is not clear. What does it mean anticipatory postural adjustments and the functional abilities within this context? The authors should better explain the latter terms.
- The terms "anticipatory postural adjustments" are written either capitalized or in lower case, and the abbreviation APA appears twice in the abstract. For consistency, I suggest not capitalize "anticipatory postural adjustments" and "virtual reality" and delete the repeated APA abbreviation.
- The abbreviation "FRT" has little correspondence with the abbreviated terms "anterior limit of stability". Reading later sections of the manuscript, I understood FRT stands for “functional reach test”. Therefore, I suggest referring to this test in the abstract or delete the abbreviation FRT.
- The sentence "... a direct impact on the anterior limit of stability..." is too general. Please specify what the "direct impact" was. Also, what is the meaning of "...patient identified by their heterogeneity"? "Patient heterogeneity" was not defined before, so its meaning is imprecise.
Introduction
Overall, the introduction is nicely written, although a bit too general. For instance, there is a lack of information regarding the specific effects of postural training on anticipatory postural adjustments and the positive outcomes of gait and balance programs using VR, as stated in the second to last paragraph of the introduction. It would be significant if the authors could be more descriptive and specific about the effects of interventions on postural control. Furthermore, the authors should present the evidence supporting their hypothesis, as stated in the introduction's concluding paragraph.
Materials and methods
Overall, the methods are sound and thoroughly presented. A few details must be clarified.
- Please indicate the exact total number of training sessions since there is confusion between 10 (based on exclusion criteria and information in figure 1) and 15 sessions (Experimental design) in three weeks.
- Please explain the reason for the 300 Hz cutoff filter for kinematic data (sampled at 100 Hz).
- The whole procedure for setting the timing of muscle recruitment is not clear to me. As I understand, t-tests were used to find the instant a particular muscle becomes active. However, how such t-tests were performed needs to be specified. Did the t-tests compare EMG baseline values with EMG values using all movement repetitions for each person and muscle? How many t-tests were performed, and how was the risk of false positives controlled?
- Muscle activation timings were "compared" by pairs. How were the correlation coefficients computed (e.g., using group average for each muscle)? We should note that correlation coefficients do not allow comparing between groups or conditions. Instead, they may inform (trough the r square) about the amount of variation in one variable, e.g., muscle activation timing in one group or condition, explained by the second variable, irrespective of the measurements being of similar magnitude or not.
- The authors should present evidence substantiating the claim for the unique role of the left erector spinae and right rectus femoris anticipatory postural adjustments.
Results
- 3B Please indicate the significant difference in finger velocity peak between the two groups at the post-test time point.
- Page 8 first paragraph, please see the above comment regarding using correlation coefficients to compare variable values. Also, although all mean activation timings occur after movement onset, the dispersion bars in Fig. 4 show anticipatory muscle activation in some movement repetitions or subjects.
Discussion
- Overall, the discussion addresses the main findings of the study. Importantly, it tries to explain the differences between the two groups in terms of the specific characteristics of the two interventions.
- Please correct "On my wonder..."
- A section addressing the study's limitations should be added to the discussion, particularly the small number of study participants and the relatively large dropout rate are two limitations that must be acknowledged.
Fig. 2 to Fig 6 do not have a legend.
Reviewer 2 Report
Imteresting manuscript that describes the use of VR in rehabilitation of geriatric patients.However i miss information about which load the patients are subjected tio during the training session as for an exmaple monitored with heart rate monitoring.
Round 2
Reviewer 1 Report
I congratulate the authors for the significantly improved revised manuscript and I thank them for the kind reply to my previous comments.
Although the manuscript has significantly improved, I still suggest the following two minor changes:
Page 9, line 244 (Figure’s 2 caption) correct m.s-1.
Page 11, line 317 & Page 13, line 353. In both sentences, it says linear correlation was used to compare the “whole activation timing sequences”. Correlations are used to assess the relationship between two variables and might be used as a measure of similarity between two sets, which seems to be the purpose of using linear correlation in this study. Therefore, I really suggest rewriting the two sentences to indicate the use of linear correlation as a measure of similarity and for comparison between groups and time points.
Author Response
Many thanks for your answer,
- The "s-1" has been corrected on the figure caption
- Line 317, we added the following sentence: "It is important to note that the linear correlations were used here as a measure of similarity and for comparison between groups and time points."
- Line 354, we added the following sentence: "As previously mentioned, linear correlations were used as a measure of similarity and for comparison between groups and time points"

Reviewer 2 Report
I am happy to note that the authors have been more specific in the load used in the train session and that they have expanded the limitation section. Nothing more to add.
Author Response
We thanks you very much for your review
Round 3
Reviewer 1 Report
Deara authors,
I am pleased to endorse your paper for publication.
Author Response
Thanks for your review